# Quantum electrodynamics at room temperature coupling a single vibrating molecule with a plasmonic nanocavity

Oluwafemi S. Ojambati[1], Rohit Chikkaraddy[1], William D. Deacon[1], Matthew Horton[1], Dean Kos[1], Vladimir A. Turek[1], Ulrich F. Keyser [1] & Jeremy J. Baumberg [1]

Interactions between a single emitter and cavity provide the archetypical system for fundamental quantum electrodynamics. Here we show that a single molecule of Atto647 aligned using DNA origami interacts coherently with a sub-wavelength plasmonic nanocavity, approaching the cooperative regime even at room temperature. Power-dependent pulsed excitation reveals Rabi oscillations, arising from the coupling of the oscillating electric field between the ground and excited states. The observed single-molecule fluorescent emission is split into two modes resulting from anti-crossing with the plasmonic mode, indicating the molecule is strongly coupled to the cavity. The second-order correlation function of the photon emission statistics is found to be pump wavelength dependent, varying from $g^{(2)}(0) = 0.4$ to 1.45, highlighting the influence of vibrational relaxation on the Jaynes-Cummings ladder. Our results show that cavity quantum electrodynamic effects can be observed in molecular systems at ambient conditions, opening significant potential for device applications.

[1] NanoPhotonics Centre, Cavendish Laboratory, Department of Physics, JJ Thompson Avenue, University of Cambridge, Cambridge CB3 0HE, UK. Correspondence and requests for materials should be addressed to J.J.B. (email: jjb12@cam.ac.uk)

Coherent interactions between single emitters and a cavity are at the heart of quantum optics, crucial both for fundamental interests in exploring the influence of quantum mechanics[1–3] and for practical applications in quantum computing, quantum cryptography, and quantum metrology[4–7]. In particular, strongly coupled systems are desirable since the reversible exchange of energy between the emitter and the cavity leads to cavity quantum electrodynamical effects[8], such as vacuum Rabi splitting[9], Rabi oscillations[1,10], non-classical photon statistics[6,11–14], and modified Purcell effects[15–17]. To achieve strong coupling, the emitter-cavity coupling rate $g$ must be greater than both cavity losses and the decoherence of the emitter[10,18]. Strong coupling has now been achieved in several systems with two-level emitters including atomic emitters, semiconductor quantum dots, vacancy centers, and two-dimensional (2D) materials inside microcavities and nanocavities[19–26].

Typically, most studies rely on cryogenically cooled emitters to achieve the strong-coupling condition, which brings multiple technical challenges and severely restricts practical implementation, scalability, and the complexity of devices. A promising alternative is to use molecular emitters since specifically-designed single molecules are easy to synthesize, can be produced on a large scale, and are integrable on-chip through careful self-assembly[27,28]. The great potential for molecular emitters lies in their identical chemical and emissive characteristics, unlike defects in TMDs, N-V centers, or quantum dots. Since the vibrational and resonance energies of a molecule can be tuned relatively easily by modifying the functional groups[29], this enables incorporation of additional functions such as reversible photochemical changes for molecular switching[30]. Even more intriguing is the capability for strong coupling with single molecules to open up the coupling of chemistry to quantum optics, and tailoring of the excited state manifold for novel control of chemical reactions.

Recent experiments with single molecules in a plasmonic cavity have shown signatures of this strong mixing of optical and exciton modes at room temperature[31–34]. Plasmonic nanocavities offer the advantage of an ultrasmall mode volume (despite their low quality factor) to achieve high coupling strengths $g \propto 1/\sqrt{V}$, where $V$ is the mode volume. Vacuum Rabi splitting which gives two spectral peaks has been observed in both scattering spectra[31–33] and emitted photoluminescence from many dye molecules assembled in J-aggregates coupled to plasmons[35,36]. Purcell effects which speed up emission and increase quantum efficiency have also been observed[37]. However, observations of strong cavity quantum electrodynamic effects that occur as a result of addressing a single molecule coupled to a nanocavity with ultrafast pulses are still lacking, and at room temperature would be a significant advance.

Here, we study the interaction of ultrafast pulses with a single molecule (Atto647) embedded within a plasmonic nanocavity formed from the nanoparticle-on-a-mirror (NPoM) geometry at room temperature. Our sample has a resonant plasmonic mode that is formed from the coupling of an 80 nm gold nanoparticle to its image charges induced in an underlying planar Au film (Fig. 1a, see SI for details of sample preparation). Between the Au nanoparticle and the Au film is sandwiched a DNA origami (DNAo) spacer which acts as a breadboard, to deterministically position a single molecule of Atto647 dye at the center of the gap (Fig. 1a, as previously proven through bleaching and positioning studies[37]). The whole sample is coated with a 30 nm polymer Parylene-C layer to shield the Atto647 from the influence of singlet oxygen, which degrades the dye over time[38]. The gap spacing between the Au nanoparticle and the Au film is determined by the DNAo spacer which is ~4.5 nm thick and has a refractive index $n \sim 2.1$[39]. Within this small gap plasmon volume ($V \sim 65$ nm$^3$) the electromagnetic field is strongly enhanced by ~170-fold, and the emission of the single molecule inside the cavity is thus significantly enhanced. Because the single-molecule emission now becomes correspondingly faster, ultrafast pulse excitation is thus required.

## Results

**Dark-field scattering**. The Atto647 molecular dye in water has an absorption spectrum maximized at 647 nm, and an emission

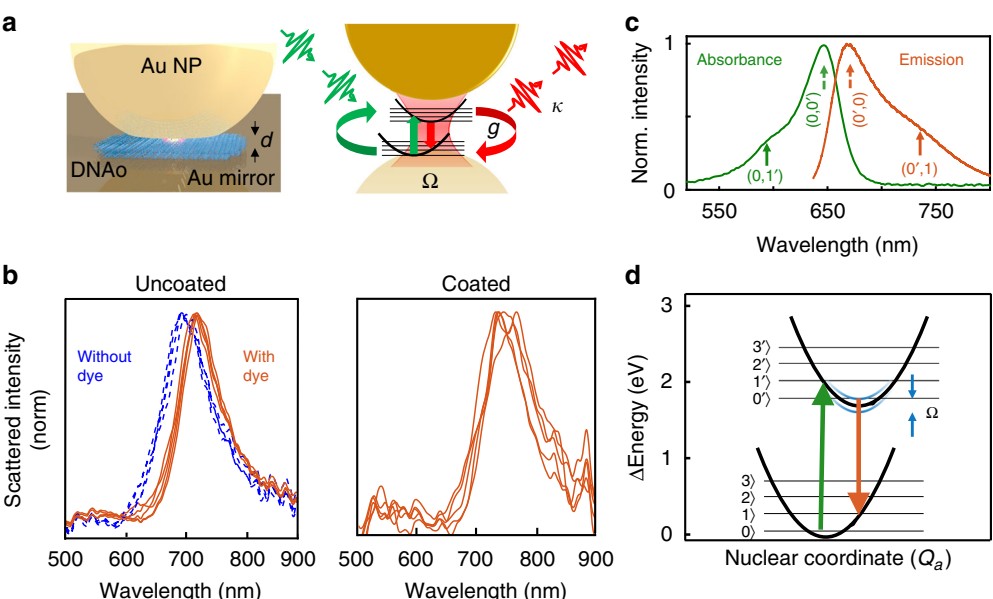

**Fig. 1** Sample and characterization. **a** Schematic of precision-located single molecule inside nanoparticle-on-mirror (NPoM) plasmonic cavity, with DNAo spacer. **b** Dark-field scattering of individual NPoMs either uncoated but (with)out molecule of Atto647, or with ParlyeneC coating. **c** Absorption and emission spectra of Atto647 dye in water. **d** Energy diagram showing electronic and vibrational energy levels of the single molecule (black), and when coupled to a strong-coupled cavity tuned to the emission wavelength (blue, width indicates light-matter coupling strength)

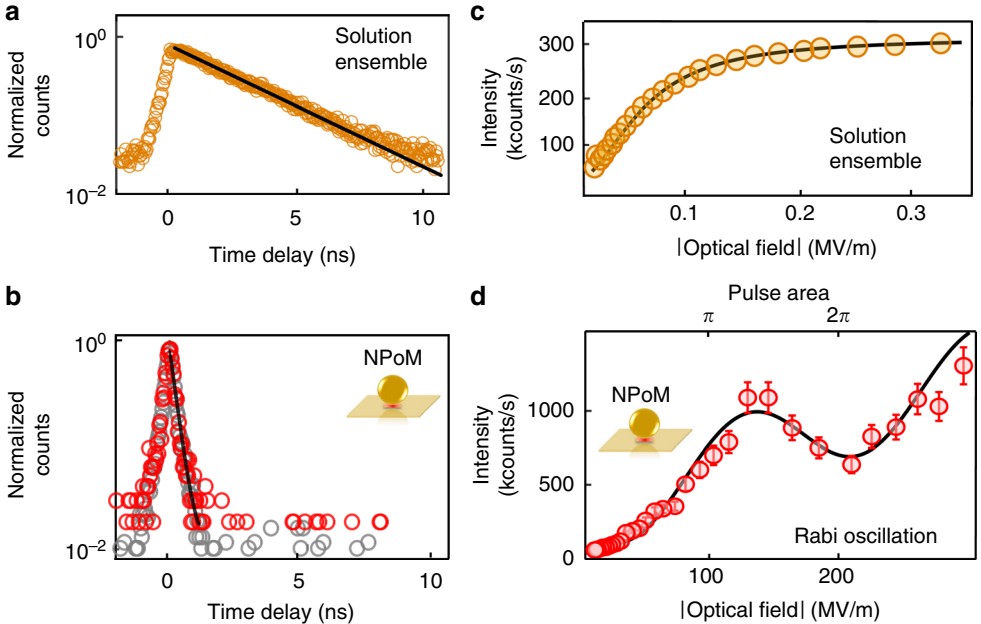

**Fig. 2** Time-resolved and power-dependent measurements. **a, b** Time-resolved emission under 120 fs pulsed excitation of **a** 10 µM concentration ensemble of Atto647 in solution (orange circles) and **b** single Atto647 in NPoM (red circles), together with instrument response (gray circles). Lines are single exponential fits. **c, d** Measured intensity vs optical field on sample for ensemble Atto647 in water and single molecules in NPoM, respectively. Curves are fits to the experimental data, described in the text. Error bars are the fluctuations of the measured counts on the detector

spectrum that peaks at 690 nm (Fig. 1b). The two spectra are mirror images, with peak and phonon sidebands well explained by the Franck–Condon effect[40] and the center and side peaks attributed to the absorption transitions $0 \rightarrow 0'$ and $0 \rightarrow 1'$. The main vibrational transition observed is at 1625 cm$^{-1}$, matching the dominant mode seen in surface-enhanced Raman spectroscopy (SERS) measurements of the same single-molecule filled plasmon cavities (Supplementary Fig. 1).

To characterize the cavities we use dark-field scattering spectroscopy[41,42] which shows a long wavelength coupled-plasmon resonance at 720 ± 20 nm without any nanoparticle coating (Fig. 1b). The presence of a single molecule of Atto647 dye in the nanocavity red-shifts the resonance by 20 nm, as expected for the strong-coupling regime (see SI S2 for further discussion). This is even further shifted to 740 ± 40 nm when the sample is now coated with a 30 nm layer of Parylene C (Fig. 1c), due to the change in refractive index around the Au NP. From the scattering spectra, this construct does not quite reach well-resolved strong-coupling compared to our recent work on Cy5[37], because the plasmon is detuned from the absorption and instead matches the emission wavelengths. Similar samples using Cy5 single molecules (with a comparable optical dipole) but smaller detuning from the absorption peak, show clear splittings in scattering spectra and yield Rabi splittings of $\Omega \sim 80$ meV[37].

**Time-resolved emission**. In this near strong-coupling regime, coherent interactions between the cavity and the single molecule are still expected, as oscillations in time domain are predicted[17]. The cavity detuning chosen here leads to splitting of the emission states, which can be modeled in the Born–Oppenheimer approximation[43,44], depicted as blue lines in (Fig. 1c). We note with single molecules that it is not possible to simultaneous operate with strong-coupling splitting at both the absorption and emission wavelengths, specifically when the Stokes shifts exceeds the Rabi splitting as here (see below). These single-molecule NPoM constructs however do allow the study of dynamic

interactions between plasmonic nanocavities with high field enhancement and a single molecule.

Comparing time-resolved emission of Atto647 in both water (at 10 µM) and for single molecules inside individual NPoMs shows the dramatic effect of the cavity (Fig. 2a, b). In each case, the samples are irradiated with 120 fs, 80 MHz repetition rate pulses tuned to 590 nm (exciting $0 \rightarrow 1'$) and the emission detected using time-correlated single-photon counting at longer wavelengths integrated across the entire emission spectrum (see Methods). The Atto647 dye in water decays following a single exponential function with lifetime $\tau_{sol} = 2.5 \pm 0.2$ ns, and in the NPoM cavity with $\tau_{NPoM} < 0.3 \pm 0.1$ ns, which is completely limited by the instrumental response (measured using attenuated pulses as 0.3 ns). The observed reduction in emission lifetime comes directly from the high density of photonic states inside the nanocavity which gives predicted Purcell enhancements >1000[17,37,45,46], as seen in the enhanced emission from these samples[37] utilized since such sub-ps single-photon times are inaccessible with current detector technologies.

**Power-dependent measurement**. Besides this emission rate speed-up, as the system approaches the strong-coupling regime oscillatory light-matter coupling is expected. Pump-driven Rabi oscillations occur in time as a periodic exchange of energy between the ground and excited states at the driven Rabi frequency $\tilde{\Omega}_R = \mu E_0 / \hbar$, where $\mu$ is the transition dipole moment and $E_0$ the amplitude of the incident field[1]. For a given pulse duration, driven Rabi oscillations are then expected in the power dependence of the emission due to changing $\tilde{\Omega}_R$ with amplitude of the incident field $E_0$. Even when the pump-induced Rabi splitting in the spectral domain is too narrow to observe, it can be seen in the power- or time domain as long as its damping is not much larger than $\tilde{\Omega}_R$.

The emitted intensity versus incident power for ensembles of Atto647 in aqueous solution (Fig. 2c) shows that saturation can be obtained beyond a linear regime and fits that expected from a

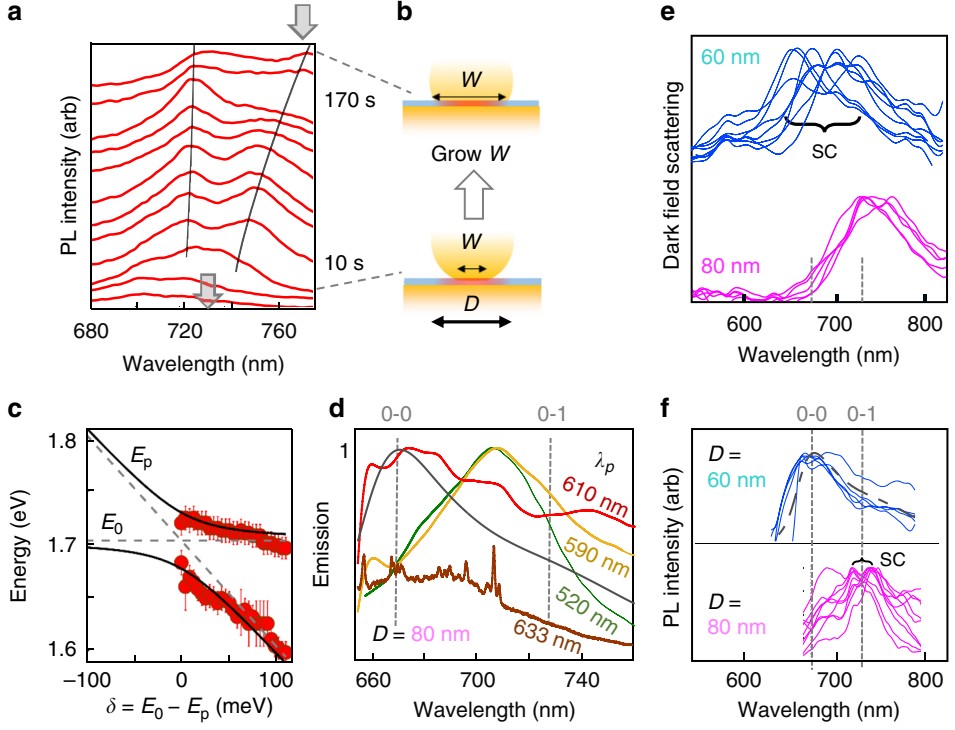

**Fig. 3** Time dependent emission. **a** Time evolution of emission spectrum of single Atto647 molecule inside plasmonic nanocavity. Spectra (offset) are at times 5, 15, 25, 55, 65, 70, 80, 90, 105, 120, 150 s, gray arrows show DF position at start and end (color map shown in Supplementary Fig. 4). **b** Schematic facet increase with time, which redshifts cavity plasmon. **c** Peak energies versus detuning $\delta = E_0 - E_p$ between plasmon ($E_p$) and dye ($E_0$) resonances. Lines are fits to Rabi coupling model. The error bar is the uncertainty of the fit to the peaks in Fig. 3a. **d** PL emission spectra for 120 fs excitation of single Atto647 molecules in $D = 80$ nm NPoM, for pump wavelengths of 520, 590, 610 nm, as well as CW excitation at 633 nm (showing SERS peaks), and CW PL spectra of dye in solution (gray dashed). **e** Dark-field scattering and (**f**) PL emission from NPoM constructs of different NP diameter, tuning the plasmon into resonance with (d) absorption or (e) emission to give strong coupling (SC)

two-level system (black curve)[47], as well as with literature[11]. With single molecular dyes inside the NPoM, it is more useful to plot the emission vs square root of power (which is proportional to field amplitude). The emission first increases quadratically with field amplitude and then shows an oscillatory behavior at high field amplitudes. This behavior fits a damped oscillatory component[48] with an additional linear term which accounts for extra background SERS from both molecule and metal[49]. This oscillatory power dependence is absent for dye molecules in solution and evidences how enhanced light-matter interactions in our molecular-plasmonic nanosystem mean that inversion can be faster than decay. We note that even though the plasmon is detuned from this $0 \rightarrow 1'$ absorption resonance, the near-field at the molecule is still strongly cavity-enhanced by ~3000 (from simulations, see Supplementary Fig. 6). Although, strong coupling is not necessary to observe pump-induced oscillations[50], the strong field confinement in the cavity is needed to overcome the competing and rapid decay processes. The oscillation has a first peak at 130 MV/m followed by a dip at 180 MV/m, corresponding to pulse areas of $1.3 \pm 0.2\pi$ and $1.8 \pm 0.2\pi$, respectively, given the cavity enhancement. These are close to the pulse areas ($\pi$ and $2\pi$), where such effects are expected. Subtracting the background contribution shows that 500 kcounts/s emerge from the single molecule when its two electronic levels are precisely inverted by an incident pulse, which is 4% efficiency compared to one photon per pulse. However the branching ratio between Rayleigh scattering ($0 \rightarrow 1' \rightarrow 0, R$), Raman ($0 \rightarrow 1' \rightarrow 1$, SERS), and luminescence ($0 \rightarrow 1' \rightarrow 0' \rightarrow 1$, PL) is not known. From our detection efficiency of 30% the real PL efficiency is 13%, and the ratio of this to the simulated emission efficiency from a dipole in the NPoM of 45% implies that the branching ratio $\frac{R}{PL} \sim \frac{45\%}{13\%} \sim 3$,

which is consistent with the enhanced Purcell Factor giving ~100 fs emission times compared to the ~0.3 ps phonon lifetimes ($1 \rightarrow 0$). We also note that the single-molecule saturation power to excite the nanocavity is several orders of magnitude larger (rather than smaller) than that in solution. This is most likely due to the lower in-coupling efficiency and rapid absorption in the Au which deexcites the molecule and is not accounted for. We further note that the Rabi oscillations are not always observed since many of the NPoM structures that we measure damage before showing noticeable oscillations. Out of 44 NPoM systems measured here, we observe a clear oscillation in over a third of them (16) (see several additional data sets in Supplementary Fig. 2).

**Rabi splitting in emission**. To further understand the molecule-cavity interaction, the emission spectrum is continuously recorded for 200 s under 120 fs pulsed pumping at 590 nm with average intensities of 5 µW/µm² (Fig. 3 and color map in Supplementary Figure 4, exposure time 5 s). For the NPoM shown here, initially the emission shows a low intensity peak (integrated ~5 kcounts/µW/s) without distinct spectral features (Fig. 3a) but after 20 s, the emitted intensity suddenly increases by a factor of four and then increases by another factor of 2 after 50 seconds, to ~40 kcounts/µW/s. Previously we have demonstrated that illumination of the NPoM system induces movement of Au facet atoms (and not simply from heating)[21,51]. Indeed, the dark-field scattering recorded at the start and end of this irradiation shows the resonance has shifted from 730 nm to 780 nm, due to the growing facet size of the nanoparticle (Fig. 3b) which red-shifts the coupled-plasmon mode. In this case the morphology change

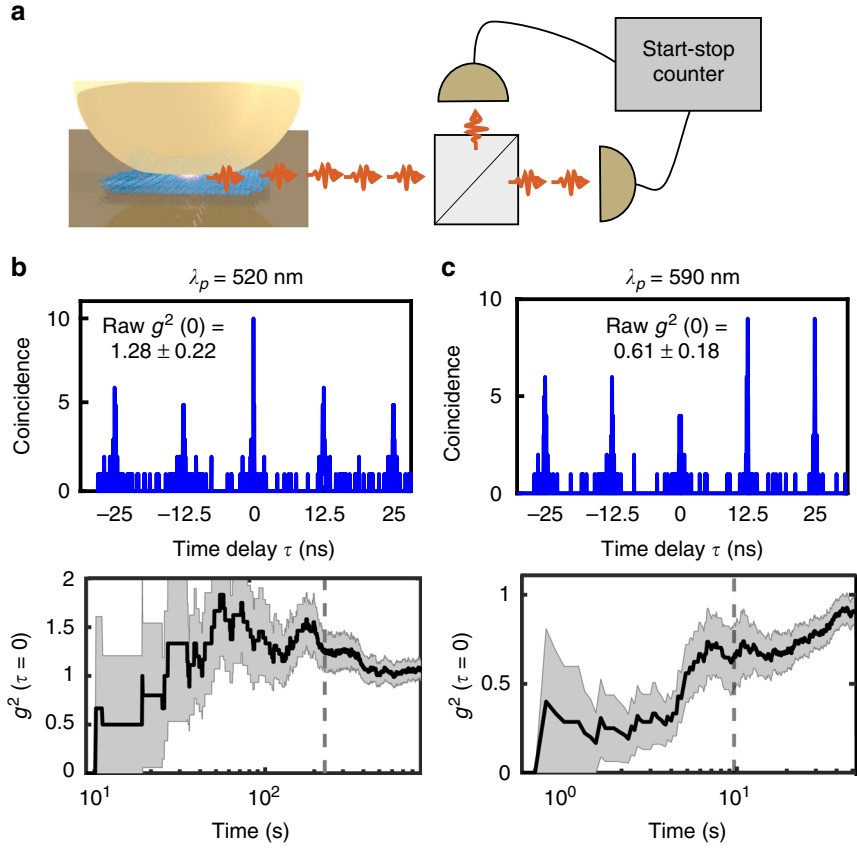

**Fig. 4** Photon correlation measurement. **a** Scheme of experiment using tunable 120 fs pulsed excitation. **b**, **c** Second-order intensity correlation of the emission light from single Atto647 molecule inside NPoM cavity using **b** 520 nm and **c** 590 nm excitation wavelengths. Bottom panels show evolution of $g^2$ (0) vs measurement time. Black dashed line shows time at which upper panels are taken

improves the molecular coupling and thus increases the photoluminescence as the local field around Au protrusions increases[52], or may reorient the dye molecule along the plasmon optical field direction (z). After this, two peaks are now apparent in the PL, red-shifted compared to the initial spectrum, and which continue to now red-shift. In other NPoMs (Supplementary Fig. 5), the two photoluminescence peaks and dark-field can remain fixed when the facets are more robust. The split PL peaks provide evidence for strong coupling in emission.

Extracting these evolving PL peak positions with time allows them to be fit to a simple vacuum Rabi coupling model based on the anticrossing of a fixed dye emission frequency with the redshifting plasmon as the cavity detuning increases (Fig. 3b, Supplementary Fig. 4)[37]. This yields a fitted single-molecule vacuum Rabi coupling strength $\Omega \approx 30 \pm 5$ meV and, since 60 meV is obtained from simulations with this vertical dipole at the center of the gap (Supplementary Fig. 6b), this suggests that the molecule is slightly off-center and/or inclined. Here the dye emission at 730 nm does not correspond to the first maximum in the solution dye luminescence spectrum (Fig. 1b) but instead is the lower energy $0' \to 1$ peak. A simple model suggests the peak emission wavelength should be found from the product of the near-field enhancement and emission spectra around 700 nm (Supplementary Fig. 6a), but this does not match the emission seen. Our data therefore suggests that the strong-coupling conditions achieved lead to changes in the branching ratio between $1' \to 0'$ and $2' \to 0'$ relaxations, and hence changes in the emission spectrum. These changes in the branching ratio, which lead to the unexpected modification of emission profile, show the complex coupling of the molecular vibronic lines to the cavity.

Although there has been some recent progress in developing models[53,54], we note that a full theory accounting for how different phonon manifolds interact with the emitting dipole in such a strong-coupling regime is currently missing. These effects can be further seen when tuning the 100 fs pump spectrum to different wavelengths (Fig. 3d) or reducing the NP size to blueshift the plasmon resonance (Fig. 3e). As discussed above, the $D = 60$ nm NPoM now brings the plasmon into absorption resonance, so that strong-coupling dark-field splittings are now observed (as before in [37]), but now the emission spectrum matches that in solution, with no strong-coupling modifications (Fig. 3f).

**Photon statistics of emission**. To investigate the statistics of the emitted photons from the single molecule in the NPoM, Hanbury–Brown–Twiss photon correlation is used (Fig. 4a, full setup in Supplementary Fig. 9). The emitted light from the molecule is split by a 50:50 beam splitter and focused onto two single-photon avalanche diodes (SPADs) with their detection times correlated electronically (resolution 200 ps). Since the dye-NPoM construct emits photons on ultrafast timescales, CW measurements of $g^2(\tau)$ are not feasible (since they average over much longer times, see SI section S6), hence 120 fs pulsed excitation at 80 MHz is adopted. Histograms of the time delay between two SPAD detection events show obvious interaction effects (Fig. 4b) as long as the excitation power is low enough to avoid damaging the molecules (<2 μW). Collection times in these experiments vary depending on photon counts and laser intensity, with collection times one order of magnitude longer at

590 nm than at 520 nm (due to lower emission from off-resonant 520 nm excitation). NPoM constructs with DNAo but with no molecule in the gap give $g^2(\tau = 0) = 1.0 \pm 0.1$ as expected from the SERS background (Supplementary Fig. 7). While excitation wavelengths of 520 nm yield bunching in second-order intensity correlations $g^2(\tau = 0) = 1.28 \pm 0.22$, when the pump is shifted to 590 nm, anti-bunching is observed, $g^2(\tau = 0) = 0.61 \pm 0.18$. The $g^2(0)$ value is calculated from the ratio of pulse areas at $\tau = 0$ to the average of areas of the other pulses at $\tau \neq 0$, i.e., $g^2(0) \equiv$
$A(\tau = 0)/\frac{1}{N}\sum_N A(\tau \neq 0)$, where $A$ is the detected pulse area[11,55]. The raw correlations then have to be corrected for the fraction of the emission which emerges not from the single molecule, but from additional background SERS-enhanced emission from the metal (extracted from Fig. 2)[51]. Correcting for this $40\% \pm 10\%$ background yields corrected second-order correlations of $g^2(0) = 1.45 \pm 0.5$ (520 nm pump) and $g^2(0) = 0.4 \pm 0.5$ (590 nm pump) (see Supplementary Note 6 for details of the correction). As the pump is tuned even closer to the peak absorption, the SERS overlaps with and becomes dominant in the emission spectra, preventing measurements of $g^2(0)$ at longer pump wavelengths (Supplementary Fig. 1). Measurements are also hindered by permanent alterations in the molecule/gold nm-scale structure which accrue, that lead to the slow changes in $g^2(0)$ with measurement time (Fig. 4b, bottom panels) arising from growth in the facet size of the nanoparticles, which can be tracked by red-shifts of the cavity resonance (Supplementary Fig. 2h, as fully detailed in[51]). The total counts observed from the single dye molecules are thus limited by the need to restrict to low pump powers (compared to Fig. 2d). The fast fluctuations in time originate from low count rates (due to the low pump power demanded) and average out at longer measurement times, or with higher signals (see Supplementary Fig. 7b).

## Discussion

Wavelength-dependent photon statistics have been observed from quantum dot (QD) systems at low temperatures and several theoretical explanations proposed[20,55–57]. They have been attributed to a combination of biexciton excitation, photon-induced tunneling, and multiple excitation of the emitter. Our observations here are the first evidence of such effects in molecular systems, which are considerably more complicated by the Stokes shifted emission process and the range of different vibrations involved (which can be ignored for QDs). We first note that the 120 fs pump pulse is not short enough to prevent multiple excitation-decay cycles within a single pulse. In QD-microcavities, short wavelength pump pulses can excite biexcitons which provide two-photon emission cascades leading to bunched light. Photon-induced tunneling can emerge from the interplay between cavity-molecule detunings and the increased Rabi splittings for increasing numbers of photons in the cavity. It is much less clear how such effects operate in the molecular system, especially since the plasmon is tuned into strong coupling with the emission here, but not the absorption. Recent theory suggests that indeed $g^2(0)$ down to 0 and exceeding 10 can be possible in molecular systems enhanced in plasmonic nanocavities[53], but however this omits the acoustic phonon bath and dephasing reservoirs. It is apparent that a full quantum analysis of this coupled electronic-vibronic system will be required, involving understanding of the absorption, emission, and plasmon tunings, together with phonon and relaxation pathways, while also exploring the effect of the close plasmonic confinement on the molecular relaxation.

In summary, we studied the interaction of ultrafast pulses with a precision-assembled single molecule that is strongly coupled to an individual ultra-low volume plasmonic nanocavity at room temperature. We observe strong Purcell effects speeding up the

dye emission, as well as driven Rabi oscillations with pump power. Splitting in the luminescence suggests the $1625\ \text{cm}^{-1}$ vibration-manifold is strongly coupled to the dominant tightly confined plasmon, producing most of the emission even at room temperature. Non-classical emission is observed, with photon bunching and anti-bunching regimes dependent on the excitation wavelength. While ubiquitous exploitation of this first demonstration is currently limited by both cavity retuning and molecular damage, methods in development aim to reduce the light-assisted motion of Au atoms within the NPoM cavity which is responsible[52]. Possible solutions for stabilization use a wider variety of ligand systems as well as encapsulation materials. This work however gives strong promise to the ability to use advanced nano-assembly techniques to produce viable single-molecule quantum emitters operational at room temperature in ambient conditions. These can pioneer many developments in the fields of quantum chemistry, nonlinear optics, and molecular quantum optics.

## Methods

**Optical setup.** The experimental setup is shown in Supplementary Fig. 9. We excite the sample with ~120fs pulses, ~10 nm full width at half maximum (FWHM), generated from a tunable optical parametric oscillator (OPO) (Spectra Physics Inspire) pumped at 820 nm with a repetition rate of 80 MHz. The power of the pulses is controlled using a linear neutral density filter (not shown) mounted on a linear translational stage. The attenuated pulses pass through a 90:10 beam splitter and are focused by a microscope objective (Nikon X100, numerical aperture 0.9) to excite the dye in the plasmonic nanocavity. Emission light passes through the beam splitter, through two long pass filters (Thorlabs FELH650, cut-off wavelength 650 nm) and is directed to a grating spectrometer using a removable mirror. The spectral image is also taken by an electron multiplying charged coupled detector (EMCCD) (Andor iXon) that is cooled to −80 °C. A typical measured emission spectrum is shown in the inset of Supplementary Fig. 9. Taking out the removable mirror directs the emission light towards a time-correlated single-photon counting (TCSPC) setup, based on two single photon avalanche diodes (SPAD) (MPD, jitter time < 50 ps) and a 50:50 plate beam splitter. Achromatic doublet lenses (focal length $f = 60$ mm) focus light onto each of the detectors. The outputs of the two SPADs are connected to a fast sampling card (Picoquant TimeHarp 200) that has a resolution of 37 ps and a jitter time of ~200 ps. The sampling card measures the time delay between the photons that are detected by the two SPADs and computes the histogram of the photon arrival time within a certain waiting time window.

## Data availability

All relevant data present in this publication can be accessed at: https://doi.org/10.17863/CAM.35395. The source data underlying Figs. 1b–c, 2a–d, 3a, c–f and 4b–c are provided as source data.

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

## Acknowledgements

We acknowledge support from EPSRC grants EP/G060649/1, EP/L027151/1, EP/G037221/1, EPSRC NanoDTC, and ERC grant LINASS 320503. O.S.O. acknowledges the support of Rubicon fellowship from the Netherlands Organisation for Scientific Research. R.C. acknowledges support from Trinity College, University of Cambridge.

## Author contributions

Experiments were planned and executed by O.S.O., W.M.D. and J.J.B. with support from M.H. Simulations were performed by R.C., while AFM measurements were performed by D.K. The samples were prepared by V.A.T., D.K. and U.F.K. The data were analyzed by O.S.O., R.C. and J.J.B., and all authors contributed to the manuscript.

## Additional information

**Competing interests:** The authors declare no competing interests.

