## [Peer Review File · Nature Communications]

Reviewers' Comments:

Reviewer #1:

Remarks to the Author:

This paper continues work of the Baumberg lab on the strong coupling of single emitters to plasmonic cavities. Here they attempt to demonstrate a few quantum optical effects on their devices, from the Purcell effect through Rabi oscillation to antibunching. However, the paper seems to be premature, since many of the data sets are not quite convincing, as detailed below. The paper also seems to be written in a hasty manner, with many details missing. In the age of the Supporting Information there is no reason for such omissions. Finally, the level of novelty is not that high.

I therefore conclude that publication in Nature Communications is not warranted, and a more technical journal might be more suitable.

Below are my specific comments:

1. The title of the paper says: "...a single vibrating molecule..." Is there a molecule that does not vibrate? There is no need to include that in the title. Incidentally, vibrations of molecules are not termed phonons. Electronic excitations that involve also vibrations are referred to as vibronic lines in the spectrum.

2. Details about the preparation of the samples should be given, rather than sending the reader to another paper of the group (as is done on the second page). Some particular issues that should be explained relate to the way the author ascertain the positioning of the molecule in the hot spot, as well as the very high index of refraction- is this for the DNA alone? With the gold nanoparticle? Etc. etc.

3. How is the addition of a single molecule shift the cavity resonance by 20 nm? Or is the spectrum of Figure 1 taken not only without the dye but also without the DNA?

4. The paper mentions a calculated Purcell effect of >1000 , but they cannot support it by their measurements. Surprisingly, they cite a 120 fs excitation pulse, but the instrument response in Figure 2b is given as 0.3 ns- 3 orders of magnitude higher! Why is that? And what is the role of dispersion in the microscope objective in broadening the pulses? And with such a broad instrument response, the discussion of why to use pulses rather than CW excitation in relation to correlation functions (below Figure 4) seems to be moot.

5. The attempt to demonstrate Rabi oscillations is confusing. First, an ensemble curve is shown that suggests saturation around a power of 100 MW per m^2 . Then a single-molecule curve with one oscillation is shown, but this oscillation appears at a power of 10^4 MW per m^2 ! Given that electric field is enhanced in the nanocavity, the oscillation should appear at a lower power than needed for saturation in the ensemble. Different numbers are given in the text, suggesting that perhaps the scale in Figure 2d was not converted to the square root scale as written. The question above remains- why such a high power is necessary for the oscillation if the field is enhanced by >1000 ?

In addition, given the intensity fluctuations reported later in Figure 3, it is not clear that the observed curve is really due to Rabi oscillations. Two examples are given, but the question is how many such curves in total were measured. Further, I don't understand why the oscillation is lost on a second run, as shown in the SI. In Figure 3 there seems to be quite a bit of intensity in the spectrum even after it is shifted, suggesting the molecule is still resonant with the cavity.

Finally, it should be mentioned that Rabi oscillations are not really a strong coupling phenomenon, and can be observed in molecules in free space, as shown by the Sandoghdar group, for example (PRA 79, 011402(R) 2009). True, they work at low temperatures, while here the experiment is done at room temperature, but the point is that strong coupling is not necessary for this observation.

6. It is not so clear what exactly happens in Figure 3a. It would be nice to see the spectra themselves. On page 4 it is stated that the construct does not quite reach well-resolved strong coupling, but then in Figure 3 it seems that a clear separation of two peaks is shown. It is

surprising, in fact, that a Rabi splitting of 30 meV is seen so well on a room-temperature emission spectrum. And why doesn't the scattering spectrum show any splitting?

7. The correlation functions in Figure 4 are corrected using a not-so-clear procedure that involves some assumptions about the contribution of SERS to the spectra. However, if SERS is strong enough to contribute, then spectra should show clear Raman lines riding on them. I don't see this in any figure. Further, in principle after correction for background the $g_2(0)$ of a single molecule should be 0- why is a larger value observed?

8. Multiple experimental and other details are missing. For example, dark field spectra in Figure 1 show a shift of 20 nm between spectra 'with dye' and 'without dye', which doesn't seem reasonable if only a single molecule is positioned in the gap. Is this also with and without the DNA origami? On page 5 a calculation of the pulse area is given without details; On the next page, first paragraph, the laser power is given rather the intensity as in other places; The collection time of the correlation functions in Figure 4b is not given, even though fluctuations of $g_2(0)$ are shown in the panels below. Finally, the experimental statistics should be included in the paper- how many devices with strongly-coupled molecules were measured? How many showed the phenomena reported in the paper?

9. Two small points: In FigS2, labelling of the panels is missing; There is no inset in Fig.3a as mentioned in the text.

Reviewer #2:

Report on Ojambati *et al.*

The submitted manuscript by Ojambati et al. reports on a room-temperature demonstration of significant spontaneous-emission enhancement of a single dye molecule due to its coupling to a plasmonic nanoparticle. Signatures of strong coupling are observed, and measurements of the autocorrelation function provide some insight into the statistical properties of the emission, even exhibiting some non-classical values.

Quantum optics and cavity-QED has largely been the domain of atomic and semiconductor physics. As the authors state in their text, their experiment merges the fields of chemistry and quantum optics by demonstrating one of the most prominent effects in cavity-QED, namely strong coupling and Purcell-enhancement, with a single molecule, which is nice and relevant.

Generally, the work may well be suited for Nature Communications. However, I believe that the impact and the application potential could be better stressed in the manuscript, and I will formulate some aspects that have remained unclear to me below. While using a dye molecule as the single quantum emitter in a cavity-QED experiment is certainly a strong argument for a high-impact publication, the results are typical and do not contain any novel physics beyond what has already been demonstrated in atomic and semiconductor systems. Additionally, the interpretation of some of the experimental results seems rather vague (see below). The editor may decide if the aspect of novelty and the expected impact on the field suffices to justify publication in Nature Communications, and I strongly suggest that the authors make stronger statements about this aspect.

In any case, I have several detailed questions and remarks that I ask the authors to address.

1)

As stated above, regarding the application potential of dye molecules in cavity-QED, I would like to see a more detailed discussion about the uncertainties reported here. The plasmonic resonance has been shown to change with excitation power, causing a significant shift of the enhanced emission energies. Furthermore, the time dependence of the autocorrelation function exhibits strong fluctuations. Together, these effects provide strong limitations to using the present emitter in NPoM geometry as a reliable light source in a desired or pre-specified emission regime.

The authors speak of significant potential for device applications, and I ask them to provide ideas how to overcome the above-mentioned limitations to bring single molecules into a device-relevant regime. What are the specific advantages promised by molecules, and how do these expectations compare e.g. to defects in atomically thin semiconductors, such as transition-metal dichalcogenides, for which strong coupling at room temperature has also been demonstrated.

2)

In Fig. 2, why are panels a) and c) referred to as “bulk”? The emission is still recorded from the single molecule, is it not?. Please clarify.

Also, I understand the argument for scaling the excitation axis proportional to the field strength in d). Why have the authors not used the same scaling in panel c), in which the x-axis scales with the intensity?

Another point regarding Fig. 2: In the main text, the oscillatory features are identified to occur roughly at π and 2π multiples of the field strength. The figure axis, on the other hand, has label 100 and 200 for the same scaling factor $\sqrt{\text{MW}/\text{m}^2}$. One of them must be wrong.

3)

I am not sure if nanocavity and plasmonic nanostructure should be used synonymously (e.g. top of page 5). In the first, the field enhancement comes from the reorganization of the photonic density of states, thereby facilitating the Purcell enhancement. In the latter case, the enhancement is due to a plasmonic resonance, which can exist even in the absence of a cavity.

4)

I am unable to follow the discussion at the end of page 5 on the branching ratio and how the estimate of $R/PL=3$ is obtained. Maybe the authors can elaborate on this.

5)

Can the authors provide an explanation why the red-shift of the resonance is observed in Fig. 3 but not in other cases? The argument made about the facet size change seems universal enough so that the effect should occur in all samples using this geometry of the plasmonic nanoparticle.

6)

The authors state that a model accounting for different phonon manifolds interacting with the emitting dipole are currently missing. The fact that the simple two-level approach does not reproduce the expected Rabi coupling strength is a sign that the model is really too simple here, is this correct? Are there any publications that have considered strong coupling in the presence of a multitude of phonon modes that could be referenced?

Also, this discussion at the turn of page 6/7 (“Our data therefore suggests...”) is quite vague. I suggest that to add a section to the Supplement containing more details if the authors believe that this is a valid and important point. Also, a simple simulation of the two decay channels could be implemented to illustrate the spectral change. I am aware that this goes beyond the current contents of the manuscript. On the other hand, quite a few statements are kept at a rather hand waving level, which in my opinion weakens the presentation of the otherwise very nice results.

7)

My greatest concern is with the evaluation of the autocorrelation data, and there are several aspects that need clarification.

- a) In section S4, g_2 is simulated by making assumptions on the photon distribution function. This is, per se, a dangerous endeavor, because nothing is known about the photon statistics, and the assumptions made could well be wrong. Moreover, even if g_2 is precisely known, the photon distribution function is not uniquely defined by it. So I wonder what benefit lies in the “simulation” when such assumptions are put into the model in the first place.
- b) While agreeing that uncorrelated photons possess a Poissonian distribution, this is definitely not necessarily the case for bunched photons. Still, this regime is modeled with a Poissonian distribution. Even for $\lambda > 1$, it reflects uncorrelated emission behavior and must yield $g_2 = 1$. It is known from textbooks, such as Loudon, that bunched photons can obey non-Poissonian distributions. Therefore, I believe that the assumptions going into the model are possibly not correct, but please prove me wrong.
- c) It is not traceable how the simulation is actually performed that leads to the results in Fig. S6. The authors should provide all the details needed to follow and to reproduce their calculation. How precisely is g_2 calculated, how is the normalization done (to the areas of the emission pulses at disconnected excitation cycles?), ...
- d) In the main text the authors state their pump pulse of 120 fs is long enough to re-excite the molecule several times during one cycle. In this context, what conclusion do the authors want to draw from their g_2 measurements? I fully agree that full quantum analysis would shed some light on this, but without a proper theoretical understanding, the insight is rather limited.
- e) In the sentence “... to the average of the rest of the $\tau \neq 0$ pulse areas.”: Do the authors really mean τ , or do they mean the next excitation cycle at a time delay T , which would then be $\tau + T$? Please clarify.

8)

The y-label is missing in Fig. S3a.

9)

Fig. 4: Why is it called “Correlation Card”? This name is unusual and not reflected in the more detailed Fig. S7.

Referee 1:

1. The title of the paper says: "...a single vibrating molecule..." Is there a molecule that does not vibrate? There is no need to include that in the title. Incidentally, vibrations of molecules are not termed phonons. Electronic excitations that involve also vibrations are referred to as vibronic lines in the spectrum.

> We believe it is important to emphasize the contribution of vibronic lines of the single molecule to the quantum optics observed, as this is different to solid state emitters. Even though every molecule vibrates, this is the first time evidence vibronic lines coupled to a cavity modify the $g(2)$.

2. Details about the preparation of the samples should be given, rather than sending the reader to another paper of the group (as is done on the second page). Some particular issues that should be explained relate to the way the author ascertain the positioning of the molecule in the hot spot, as well as the very high index of refraction- is this for the DNA alone? With the gold nanoparticle? etc.

> We concur that it is helpful to include more details about the sample preparation in this manuscript. We include a paragraph on page 2 as well as further specifics in the Supp.Info. We have recently shown how the DNA-tagged dye molecules can be inserted into DNA origami to give prescriptive positions within 2nm (see [37]), and we now discuss this. The refractive index of the DNA origami comes from previous papers [37,39,41] which extract it from careful fitting of coupled mode wavelengths (ie. when in close proximity to Au surfaces), however there have been many discussions on why it might be so high. We now also comment on this in the SI.

3. How is the addition of a single molecule shift the cavity resonance by 20 nm? Or is the spectrum of Figure 1 taken not only without the dye but also without the DNA?

> We confirm that in Figure 1 the curve without dye still has the DNA in place, now noted in the caption. This redshift is indeed an interesting observation that may arise from several sources. The resonant refractive index of the molecule can indeed shift of the cavity resonance. Red-shifts of > 50 nm have been observed for tens of non-resonant molecules [41]. With a mode area here of $\sim Rd/n^2$ (for NP radius R , gap d , and gap refractive index n , see [37,41]), the plasmon mode covers the area of ~ 40 dye molecules, which have resonant refractive index of ~ 4 [Cacciola *et al*, ACS Photonics **2**, 971 (2015), Plekhanov *et al*, Opt.Spectrosc. **104**, 545 (2008)]. Estimating (see Supp.Info.) the resulting shift from this small change in refractive index plausibly explains the observed 2% redshift. In the scenario of strong coupling, the shift comes from the negatively detuned lower polariton, arising similarly to above from the phase shift induced during a single round trip by the resonant molecule in the cavity dispersion. In similar samples [35], indeed the splitting from such single molecules is observed. The discussion added to the main text and Supp.Info. clarifies this in much more detail.

4. The paper mentions a calculated Purcell effect of >1000, but they cannot support it by their measurements. Surprisingly, they cite a 120 fs excitation pulse, but the instrument response in Figure 2b is given as 0.3 ns- 3 orders of magnitude higher! Why is that? And what is the role of dispersion in the microscope objective in broadening the pulses? And with such a broad instrument response, the discussion of why to use pulses rather than CW excitation in relation to correlation functions (below Figure 4) seems to be moot.

> The quoted Purcell factor >1000 is already demonstrated from enhanced emission experiments for these samples in [37] (so not repeated here), and supported by theory calculations in [17]. We stress that no single-photon detector is yet capable of 100fs response, and the SPADs used here have an instrumental detection response of 0.3ns as typical. We use autocorrelation at the sample to confirm that the microscope objective has little broadening effect on the pulse (<200fs FWHM optical response).

This discussion shows why CW excitation cannot be used. The CW $g^2(0)$ would show a deep dip with a sub-100fs recovery time. However since any single-photon detection averages over 0.3ns, then the dip would be electronically smoothed out by 0.3ns/100fs~3000 times, making the dip now 3000x broader and 3000x smaller, impossible to see over the typical noise. We stress that no group has ever managed such experiments, and in our view they are not feasible (though perhaps the referee has an idea?). Instead then pulsed excitation has to be used, with spacings in time larger than this instrumental time resolution. The 100fs excitation needs to match the emission time (which is <1ps from the Purcell enhancements measured in [37]) to avoid reexcitation of the dye within a single pulse. This useful discussion is now further clarified on pages 3 and 5 and in the SI Section S6.

5. The attempt to demonstrate Rabi oscillations is confusing. First, an ensemble curve is shown that suggests saturation around a power of 100 MW per m². Then a single-molecule curve with one oscillation is shown, but this oscillation appears at a power of 10⁴ MW per m²! Given that electric field is enhanced in the nanocavity, the oscillation should appear at a lower power than needed for saturation in the ensemble. Different numbers are given in the text, suggesting that perhaps the scale in Figure 2d was not converted to the square root scale as written. The question above remains- why such a high power is necessary for the oscillation if the field is enhanced by >1000?

> We have carefully checked all the powers. The reviewer wonders why the power at which the Rabi oscillation occurs in the NPoM is higher than the power at which saturation occurs in the dye solution. With pulsed excitation, a single molecule inside the cavity is indeed expected to saturate at $E/E_0 \sim 170$ times less power (due to the field enhancement in the cavity) than the pump power needed for molecules in solution. The oscillation effect observed is at $\sim 60 \sqrt{(\text{MWm}^{-2})}$, two orders of magnitude higher than in solution [where our data agree with prior work, for instance *Nature Methods* 4, 81 (2007)]. This difference can be attributed to a combined contribution of a low coupling efficiency (~10%) and a >30% de-excitation from photon absorption into the AuNP, and we discuss this further now in the text. The reviewer also helpfully notes that the numbers in the text are different from the x-axis of Fig. 2d, and we now corrected these numbers in the text.

In addition, given the intensity fluctuations reported later in Figure 3, it is not clear that the observed curve is really due to Rabi oscillations. Two examples are given, but the question is how many such curves in total were measured. Further, I don't understand why the oscillation is lost on a second run, as shown in the SI. In Figure 3 there seems to be quite a bit of intensity in the spectrum even after it is shifted, suggesting the molecule is still resonant with the cavity.

> The referee asks how many samples show such Rabi oscillations. We find about 40% of DNAo single-dye NPoMs show this behaviour, and believe that the molecule is often damaged before the oscillation is seen. The reason oscillations can be lost on the second run is that the plasmon cavity mode is shifted by the movement of Au atoms (see [49,51]) and although some emission is seen, it is no longer in resonance, as clearly observed in Fig.3a. We note that there is no alternative explanation that might explain this oscillation. We now clarified this further in the SI.

Finally, it should be mentioned that Rabi oscillations are not really a strong coupling phenomenon, and can be observed in molecules in free space, as shown by the Sandoghdar group, for example (PRA 79, 011402(R) 2009). True, they work at low temperatures, while here the experiment is done at room temperature, but the point is that strong coupling is not necessary for this observation.

> We fully agree that strong coupling is not needed to observe pump-induced Rabi oscillations in absorption, but this is not what we claim. Instead it confirms that control over this 2-level system in the plasmonic cavity is achieved. We clarify this point more fully in the text as suggested, citing also

the noted PRA paper. We carefully distinguish now between pump-driven Rabi oscillations, and vacuum-Rabi splitting throughout.

6. It is not so clear what exactly happens in Figure 3a. It would be nice to see the spectra themselves. On page 4 it is stated that the construct does not quite reach well-resolved strong coupling, but then in Figure 3 it seems that a clear separation of two peaks is shown. It is surprising, in fact, that a Rabi splitting of 30 meV is seen so well on a room-temperature emission spectrum. And why doesn't the scattering spectrum show any splitting?

> As requested, we now include these spectra in Fig. 3a in order to clearly show the evolution of the measured photoluminescence. The reviewer remarks that the two clear peaks indicate strong coupling but our text is more cautious, for exactly the reason that while two emission peaks are seen the splitting only marginally exceeds thermal broadening. This is however an important advance since it has not been previously possible to see two split peaks in emission from single-molecule strong-coupling (feedback from showing our data at meetings has made clear that the community see this as a very significant finding). As a result, we now stress this in the manuscript. We do not see two clear peaks split in the scattering spectra because the plasmon resonance is detuned from the absorption peak of the emitter. This emphasises how careful one has to be looking at molecules in plasmonic strong coupling: resonance can only be achieved at either the absorption or emission wavelengths, but not both. We now add further data in Fig.3 showing the effect of tuning the plasmon resonance, which emphasises this point, highlighting the differences to semiconductor or atomic systems.

7. The correlation functions in Figure 4 are corrected using a not-so-clear procedure that involves some assumptions about the contribution of SERS to the spectra. However, if SERS is strong enough to contribute, then spectra should show clear Raman lines riding on them. I don't see this in any figure. Further, in principle after correction for background the $g^2(0)$ of a single molecule should be 0- why is a larger value observed?

> As suggested, we have rewritten Section S4 of the SI to more clearly explain the simulation that accounts for background contributions. The contributions to the spectra are not assumed but measured. The reviewer also wonders why SERS peaks are not visible on the measured spectra. The reason is that the SERS peaks are at shorter wavelengths (close to the excitation wavelength) and outside our detection range – with a longer wavelength pump they appear and dominate making it impossible to separate SERS from emission (see additional data provided in Fig. S1). The reviewer asks why $g^2(0) > 0$, even after correcting for this background. We mention in the SI that multiple excitation cycles of the molecule during one pulse can give this result, as the emission lifetime is <100fs. We now discuss this further in Section S4 of the SI. We also add the $g^2(0)$ data for NPoM constructs with the DNAo but without the single molecule attached inside (Fig.S6).

8. Multiple experimental and other details are missing. For example, dark field spectra in Figure 1 show a shift of 20 nm between spectra 'with dye' and 'without dye', which doesn't seem reasonable if only a single molecule is positioned in the gap. Is this also with and without the DNA origami? On page 5 a calculation of the pulse area is given without details; On the next page, first paragraph, the laser power is given rather the intensity as in other places; The collection time of the correlation functions in Figure 4b is not given, even though fluctuations of $g^2(0)$ are shown in the panels below. Finally, the experimental statistics should be included in the paper- how many devices with strongly-coupled molecules were measured? How many showed the phenomena reported in the paper?

> These are all helpful to include as suggested. The question about the 20nm shift is repeated from their point 3, where we answer it in detail. We provide more details now for the calculation of the pulse area in the SI. We also use laser intensity throughout. The collection times for $g^2(0)$ are reported,

and the experimental statistics discussed in detail. We have detailed measurement on tens of NPoMs, ranging over different detunings, however as we note in the manuscript they still ultimately damage preventing many repeated measurements.

9. Two small points: In FigS2, labelling of the panels is missing; There is no inset in Fig.3a as mentioned in the text.

> We thank the referee for spotting these and have corrected them appropriately.

Referee 2:

1) As stated above, regarding the application potential of dye molecules in cavity-QED, I would like to see a more detailed discussion about the uncertainties reported here. The plasmonic resonance has been shown to change with excitation power, causing a significant shift of the enhanced emission energies. Furthermore, the time dependence of the autocorrelation function exhibits strong fluctuations. Together, these effects provide strong limitations to using the present emitter in NPoM geometry as a reliable light source in a desired or pre-specified emission regime. The authors speak of significant potential for device applications, and I ask them to provide ideas how to overcome the above-mentioned limitations to bring single molecules into a device-relevant regime. What are the specific advantages promised by molecules, and how do these expectations compare e.g. to defects in atomically thin semiconductors, such as transition-metal dichalcogenides, for which strong coupling at room temperature has also been demonstrated.

> Indeed the promise of these room temperature devices, also highlights some current problems which is natural since this is the first demonstration of such capabilities. We do not believe that it is realistic to ask for solutions in this paper, but agree it is sensible to discuss them. The degradation effects are related to the movement of Au atoms which is still under active investigation, and different ligand systems are now available that seem to stabilise this, as well as different varieties of encapsulation. We now highlight this discussion in the conclusion.

2) In Fig. 2, why are panels a) and c) referred to as “bulk”? The emission is still recorded from the single molecule, is it not?. Please clarify. Also, I understand the argument for scaling the excitation axis proportional to the field strength in d). Why have the authors not used the same scaling in panel c), in which the x-axis scales with the intensity? Another point regarding Fig. 2: In the main text, the oscillatory features are identified to occur roughly at π and 2π multiples of the field strength. The figure axis, on the other hand, has label 100 and 200 for the same scaling factor $\sqrt{\text{MW}/\text{m}^2}$. One of them must be wrong.

The measurements in Fig.2(a,c) are actually performed on ensembles of molecules in aqueous solution so we now change the labels to “solution ensemble”. As suggested, we change the x-axis in panel (c) to field strength as in panel (d), which also helps the comparison requested by referee 1. As discussed for referee 1, we clarify and carefully correct the powers and pulse areas on these graphs.

3) I am not sure if nanocavity and plasmonic nanostructure should be used synonymously (e.g. top of page 5). In the first, the field enhancement comes from the reorganization of the photonic density of states, thereby facilitating the Purcell enhancement. In the latter case, the enhancement is due to a plasmonic resonance, which can exist even in the absence of a cavity.

The terms ‘nanocavity’ and ‘plasmonic nanostructure’ mainly differ in whether they consider closed or open modes (respectively). However the trapped mode in this NPoM structure is definitely a type of Fabry-Perot metal-insulator-metal waveguide [see ACS Nano 9, 825; PRA 92, 053811] and thus a type of ‘plasmonic nanocavity’, the term which we now use consistently throughout.

4) I am unable to follow the discussion at the end of page 5 on the branching ratio and how the estimate of $R/PL=3$ is obtained. Maybe the authors can elaborate on this.

We observe a raw PL efficiency of 0.04, which with detection efficiency of 0.3 gives actual PL efficiency of 0.13. Our simulations show a dipole emission efficiency should be 0.45. The reason less is obtained is that the excited molecule can either emit a PL photon, or a Rayleigh elastically scattered photon, so the branching ratio between these is $0.45/0.13 \sim 3$. We added this to the text.

5) Can the authors provide an explanation why the red-shift of the resonance is observed in Fig. 3 but not in other cases? The argument made about the facet size change seems universal enough so that the effect should occur in all samples using this geometry of the plasmonic nanoparticle.

We generally see different thresholds for light-induced red-shifts on different particles, likely due to different facet sizes, geometries and adatom energies at the facet edges, see [49]. We now discuss this explicitly in the SI.

6) The authors state that a model accounting for different phonon manifolds interacting with the emitting dipole are currently missing. The fact that the simple two-level approach does not reproduce the expected Rabi coupling strength is a sign that the model is really too simple here, is this correct? Are there any publications that have considered strong coupling in the presence of a multitude of phonon modes that could be referenced? Also, this discussion at the turn of page 6/7 (“Our data therefore suggests...”) is quite vague. I suggest that to add a section to the Supplement containing more details if the authors believe that this is a valid and important point. Also, a simple simulation of the two decay channels could be implemented to illustrate the spectral change. I am aware that this goes beyond the current contents of the manuscript. On the other hand, quite a few statements are kept at a rather hand waving level, which in my opinion weakens the presentation of the otherwise very nice results.

The referee is correct that this is a key point of the paper, that the simple model is unable to predict the results because it is very hard to calculate g^2 in the presence of strong coupling and phonons. We have been working with a range of collaborators who are developing complex theoretical models to handle the g^2 in this situation. But so far these are too preliminary to explain more than that indeed $g^2 < 0.5$ is expected, as would be from any quantum emitter. We thus reference other theory works [53,54], and believe our work will act as a strong stimulus for the theory community, who see such ultrasmall volume cavities as providing dramatic new hierarchies of coherence and energy scales. We are aware of now 7 theoretical groups now working extremely hard on this problem, making this paper all the more timely as it provides them the key data for their simulations.

7) My greatest concern is with the evaluation of the autocorrelation data, and there are several aspects that need clarification.

a) In section S4, g^2 is simulated by making assumptions on the photon distribution function. This is, per se, a dangerous endeavor, because nothing is known about the photon statistics, and the assumptions made could well be wrong. Moreover, even if g^2 is precisely known, the photon distribution function is not uniquely defined by it. So I wonder what benefit lies in the “simulation” when such assumptions are put into the model in the first place.

We present the raw data here, as well as a simple model to correct for background contributions, thus giving all information to the reader. The correction is based on single photon emission plus a Poissonian background which comes from electronic Raman processes, which is assumed to be classical. This is a good assumption since the electron bath is involved from the metal (ie not a single excitation), but we now note this explicitly.

b) While agreeing that uncorrelated photons possess a Poissonian distribution, this is definitely not necessarily the case for bunched photons. Still, this regime is modeled with a Poissonian distribution. Even for $\lambda > 1$, it reflects uncorrelated emission behavior and must yield $g_2 = 1$.

It is known from textbooks, such as Loudon, that bunched photons can obey non-Poissonian distributions. Therefore, I believe that the assumptions going into the model are possibly not correct, but please prove me wrong.

Indeed, Poissonian distributions describe uncorrelated photons, which is not the case for bunched photons. Bunched photons (e.g. thermal or chaotic light) can be described by a super-Poissonian distribution that has a variance greater than the mean. We use the example of the Bose-Einstein distribution [1,48] which we have implemented, and while it is not a unique choice it is the one frequently used. We provide a more complete discussion now in Section S4.

c) It is not traceable how the simulation is actually performed that leads to the results in Fig. S6. The authors should provide all the details needed to follow and to reproduce their calculation. How precisely is g_2 calculated, how is the normalization done (to the areas of the emission pulses at disconnected excitation cycles?), ...

As suggested, we now explicitly include all details of the simulation in SI Section S4 with brief details added in the calculation of g_2 on page 8, paragraph 1 of the main paper.

d) In the main text the authors state their pump pulse of 120 fs is long enough to re-excite the molecule several times during one cycle. In this context, what conclusion do the authors want to draw from their g_2 measurements? I fully agree that full quantum analysis would shed some light on this, but without a proper theoretical understanding, the insight is rather limited.

Our statement simply explains why it is possible to have $g_2(0) > 0$ when corrected, as even then a single-photon source can emit two photons per pulse. It comes from the unprecedented fast emission rate of the molecule in this ultrasmall plasmonic cavity, hence reexcitation within a single pulse is possible. However as we discuss, other origins are also candidates, and a full theory is needed to tackle these.

e) In the sentence "... to the average of the rest of the $\tau \neq 0$ pulse areas.": Do the authors really mean τ , or do they mean the next excitation cycle at a time delay T , which would then be $\tau + T$? Please clarify.

Indeed our terminology was to compare to the pulses at $\tau + nT$ with $n = \pm 1, \pm 2, \dots$ and we thus compare not to the next excitation, but all the $n \neq 0$ pulses. We clarify this now.

8) The y-label is missing in Fig. S3a.

This has been fixed now.

9) Fig. 4: Why is it called "Correlation Card"? This name is unusual and not reflected in the more detailed Fig. S7.

We re-label this diagram, as a 'start-stop counter' to reflect its operation.

Reviewers' Comments:

Reviewer #1:

Remarks to the Author:

The authors have addressed some of the points I raised in my review satisfactorily. Unfortunately, some points remain only partially answered, and I am worried about the level of rigor of the data and analysis. I therefore still find the paper way too premature for publication in Nature Communications. Below are my detailed comments, numbered as in the rebuttal letter.

1. "this is the first time evidence vibronic lines coupled to a cavity modify the $g(2)$ "- There is no such evidence in the paper. The fact that the $g2$ function gives a different value at 0 time when the excitation is shifted to 520 nm cannot be taken as evidence for vibronic involvement. In fact, looking at the $g2$ functions and their evolution with time, one cannot even conclude clearly that there are differences between different excitation wavelengths. Even more worrisome is the fact that this measurement fluctuates so much over time, as this calls into question the assertion of the authors that the position of the molecule within the cavity is stable and well-determined, as suggested e.g. in their previous paper (ref. 37).

3. If the plasmon mode covers ~40 molecules, as now pointed by the authors, how do they claim that a single molecule is responsible for the signal?

4. The time resolution in a TCSPC experiment is not determined by the detector but by the electronics, usually easily reaching picoseconds. Indeed, the authors quote a resolution of 37 ps in the paper. So the broad instrument response function still remains a mystery.

5. I don't see how issues of coupling and absorption in the nanoparticles can lead to the requirement for two orders of magnitude more intense excitation power. I wonder if there is clear evidence that at this laser power there is no significant local heating and even structural changes in the particles and the surface. What happens to the device over time with this amount of light is a question that requires a clear experimental answer.

The authors evade my question about the number of repetitions of the Rabi experiment by stating that they saw the phenomena in 40% of the devices. Since I'm sure they did not do a power sweeping experiment on every device measured, this answer cannot be correct. Can the number of repeats of this experiment be clearly stated, as well as how many of them showed the 'Rabi oscillation'? In the absence of such a number, I conclude that it was seen only twice. As I noted in my previous comments, with the level of noise and signal fluctuations shown e.g. in the $g2$ experiment or in Figure 3, it is not clear that they really see a Rabi oscillation.

Reviewer #2:

None

NCOMMS-18-17853: “Quantum electrodynamics at room temperature ...”

Response to referees:

We are delighted that referee 2 now recommends publication in Nature Communications, while the additional useful questions of referee 1 are answered below and give suitable clarifications to help the reader.

Referee 1:

1. “this is the first time evidence vibronic lines coupled to a cavity modify the g^2 ” - There is no such evidence in the paper. The fact that the g^2 function gives a different value at 0 time when the excitation is shifted to 520 nm cannot be taken as evidence for vibronic involvement. In fact, looking at the g^2 functions and their evolution with time, one cannot even conclude clearly that there are differences between different excitation wavelengths. Even more worrisome is the fact that this measurement fluctuates so much over time, as this calls into question the assertion of the authors that the position of the molecule within the cavity is stable and well-determined, as suggested e.g. in their previous paper (ref. 37).

> Indeed no such statement is made in the paper. All we say here is that “*Wavelength-dependent photon statistics have been observed from quantum dot systems at low temperatures and several theoretical explanations proposed... Our observations here are the first evidence of such effects in molecular systems*” (p10) and this is fully evidenced in our paper. We agree with the referee that the observations we show of $g^2(0)$ at different wavelengths cannot be yet conclusive, without more developed theories that we exactly aim to stimulate. However it certainly appears that vibronic coupling is involved since it clearly modifies the emission spectrum of the molecule in the cavity.

The slow changes in $g^2(0)$ result from growth in the facet size of the nanoparticles, which red-shifts the cavity resonance (as fully detailed in [51]). The fast fluctuations in time originate from the low counts and average out at longer measurement time, or with higher signals (see Fig. S7b).

Previous measurements in [37] and [17] give full confidence that the position of the molecule is fixed (see point 2 below). The fluctuations in intensity seen in Fig. 3(a) are caused by the movement of Au adatoms, see [52]. We now emphasise these points in the revised manuscript.

2. If the plasmon mode covers ~40 molecules, as now pointed by the authors, how do they claim that a single molecule is responsible for the signal?

> In our previous reply to the referee, we were not clear enough when we said “*Red-shifts of > 50 nm have been observed for tens of non-resonant molecules [41]*”. Our calculation confirming that the spectral shift can be produced by a single dye molecule does not depend on this, but only on the mode area which [37] reveals precisely and matches theory. Indeed, the plasmon mode area is ~40 times larger than the area of a single molecule. In [37] we also confirm these constructs can only have a single dye molecule (there is only a single binding site on each DNAo, and we show clear single molecule bleaching in this work).

3. The time resolution in a TCSPC experiment is not determined by the detector but by the electronics, usually easily reaching picoseconds. Indeed, the authors quote a resolution of 37 ps in the paper. So the broad instrument response function still remains a mystery.

> The referee suggests no way to measure such sub-ps correlations. Our TCSPC instrument response is limited to ~150 ps by the time uncertainty of the Time-to-Digital conversion of our correlation card (Picoquant TimeHarp 200). While faster correlation cards exist, these are none of them fast enough to resolve the ultrafast time decay of emission in these cavities (<100fs), so this comment is not relevant.

4. I don't see how issues of coupling and absorption in the nanoparticles can lead to the requirement for two orders of magnitude more intense excitation power. I wonder if there is clear evidence that at

this laser power there is no significant local heating and even structural changes in the particles and the surface. What happens to the device over time with this amount of light is a question that requires a clear experimental answer.

The authors evade my question about the number of repetitions of the Rabi experiment by stating that they saw the phenomena in 40% of the devices. Since I'm sure they did not do a power sweeping experiment on every device measured, this answer cannot be correct. Can the number of repeats of this experiment be clearly stated, as well as how many of them showed the 'Rabi oscillation'? In the absence of such a number, I conclude that it was seen only twice. As I noted in my previous comments, with the level of noise and signal fluctuations shown e.g. in the g2 experiment or in Figure 3, it is not clear that they really see a Rabi oscillation.

> Indeed as we state, the cavity undergoes tiny facet structural changes during the power series measurement which gives red-shifting cavity resonances (detuned from the molecule emission). The dominant nonlinearity we have shown over the past few years is thus from the Au atoms in the gap. Evidence of this is shown in dark-field scattering plots that we now include in the supplementary material (Fig.S2h, Fig.S3), that shows the red-shifted plasmonic cavity mode. This structural change is however minimal for the first power sweep, since the sample is only illuminated for 5 seconds at each power step. We show more evidence of this in Fig.S3 which shows that instead of the molecule's bleaching, the redshift of the plasmon reduces the overlap with the pump excitation and the overlap of the outgoing PL with the plasmon, so gradually reducing the PL emission.

As the referee points out, the Rabi oscillations are quite hard to observe since most of the samples that we measured have damaged before showing noticeable oscillations. Out of 44 NPoM systems that we measured, we observed a clear oscillation in over a third of them (16). We now discuss this in the text, and show an additional few data sets in Fig.S2(e-g).

Finally we emphasise that at such average powers of $1-5 \mu\text{W}/\mu\text{m}^2$, heating effects are not observed (we see them typically at $>100 \mu\text{W}/\mu\text{m}^2$ [see Fig.8 in PRX **8**, 011016 (2018)]). We also are able to show at these pulsed power levels (Fig.15b of the same paper) that the temperature (as measured from the anti-Stokes background) remains at room temperature. Hence the typical molecular bond-breaking processes do not seem to operate in the power regime used here. We now add a discussion of this to the paper.

With these changes, we believe we improve the manuscript, answering all the referee's comments, and that this work is suited for publication in Nature Communications. We are certainly aware that a number of groups in the community ask us for these results consistently, and there is a high level of interest in this research.

Reviewers' Comments:

Reviewer #1:

Remarks to the Author:

Although I still have some reservations about the results, I think that to a large extent the authors have answered my questions, and the paper should be published so that the scientific community can see the results and judge for itself.

Reviewer #2:

None